

# The "Ocean Carbon States" Database: a proof-of-concept application of cluster analysis in the ocean carbon cycle

Rebecca Latto[1,2], Anastasia Romanou[1,2]

[1]Applied Physics and Applied Math, Columbia University, New York, USA
[2]NASA-GISS, New York, NY, USA

*Correspondence to*: rl2797@columbia.edu

**Abstract.** In this paper, we present a database of the basic regimes of carbon cycle in the ocean as obtained using data mining and patter recognition techniques in observational as well as model data. Advanced data mining techniques are becoming widely used in Climate and Earth Sciences with the purpose of extracting new meaningful information from large and complex datasets. Such techniques need to be rigorously tested, however, in simple, well-understood cases to better assess their utility. This is particularly important for studies of the global carbon cycle, where the interaction of physical and biogeochemical drivers confounds our ability to accurately describe, understand, and predict $CO_2$ concentrations and their changes in the major planetary carbon reservoirs. In addition to observational data of the carbon cycle, numerical simulations of the Earth System are becoming increasingly more complex and harder to evaluate. Without reliable numerical models, however, our predictions of future climate change are haphazard. Here we describe the use of a specific data-mining technique, cluster analysis, as a means of identifying and comparing spatial and temporal patterns extracted from observational and model datasets. As the observational data is organized into various regimes, which we will call "ocean carbon states", we gain insight into the physical and/or biogeochemical processes controlling the ocean carbon cycle in nature as well as how well these processes are simulated by the model. Assessment of cluster analysis results demonstrates that this technique effectively produces realistic, dynamic regimes that can be associated with specific processes at different temporal scales for both observations and the model. Furthermore, these regimes can be used to illustrate and characterize the model biases in the model air-sea flux of $CO_2$ which are then attributed to model misrepresentations of salinity, sea surface temperature, wind speed, and nitrate. The goal of this proof-of-concept study is to establish a methodology for implementing and interpreting k-means cluster analysis on observations and model output which will enable us to subsequently apply the analysis to larger, higher frequency datasets of the ocean carbon cycle. To enable further testing and extending the method discussed, all data and analysis scripts are freely available at data.giss.nasa.gov/oceans/carbonstates/ (DOI: 10.5281/zenodo.996891)

## 1 Introduction

Global warming, i.e. the rising temperatures in the Earth's troposphere, results from increasing concentrations of the atmospheric airborne fraction of $CO_2$, alongside other greenhouse gases. The evolution of this airborne fraction depends on



the behavior of the land and ocean carbon sinks. Here we will address the latter and specifically the ocean carbon representation in observations and numerical model simulations of Earth's climate. We aim to develop a methodology for defining the basic regimes, or "ocean carbon states", that describe its variability.

A key process in the ocean carbon cycle is the air-sea exchange of $CO_2$ (Sarmiento and Gruber 2006; Takahashi et al., 2009) that is described by Eq. (1):

$$F = kw \; K_0 \, (pCO_{2atm} - pCO_{2sw}) \tag{1}$$

where kw is the piston velocity for $CO_2$ (in ms$^{-1}$) that depends on the wind speed, $K_0$ is the solubility coefficient- dependent on sea surface temperature (SST) and sea surface salinity (SSS) (expressed in mole,$CO_2$kg$^{-1}$atm$^{-1}$) and $pCO_2$ is the partial

pressure of $CO_2$ (Wanninkhof et al., 2013) in the atmosphere (atm) and the surface ocean (sw). Eq. (1) describes the chemical disequilibrium of $CO_2$ in the oceanic and atmospheric reservoirs due to the solubility and biological pumps. As discussed in Sarmiento and Gruber (2006), the $pCO_{2sw}$ in Eq. (1) is a function of temperature and salinity, wind speed, dissolved inorganic carbon (DIC = the sum of all inorganic carbon species), nutrients, and alkalinity (a measure of the excess of bases over acids) which can be expressed as follows:

$$pCO2(ocean) = f(SST, SSS, DIC, windspeed, nutrients, alkalinity) \tag{2}$$

In this study and in the numerical simulations used, alkalinity is assumed analogous to surface salinity, which is an acceptable approximation for the sea surface but does not take into account changes in the carbonate pump. Temperature and salinity are affected only by physical processes such as circulation, advection, eddy mixing and stirring, and local upwelling/downwelling

while DIC distributions are influenced by all these physical processes and also several biogeochemical processes such as air-sea gas exchange, production by organisms, biological export to depth and remineralization there and nutrient availability in the water column

Eq. (1) points to two main pathways that determine the ability of the ocean to take up $CO_2$: the chemical disequilibrium

expressed by $pCO_2$, DIC, and nutrients, and the physical processes, such as air-sea interaction (expressed by the wind speed) and ocean circulation (expressed by sea surface temperature and salinity). Greater insight into the ocean's biogeochemical processes that control these pathways can inform the improved use of field measurements, the development of better metrics for model evaluation, and the selection of more suitable parameterizations in climate models in order to provide more accurate predictions. We select a pair of independent variables , $pCO_{2sw}$ and SST, that are able to represent a broad range of

biogeochemical and physical processes and use them in cluster analysis to find temporal and spatial patterns in their joint parameter space that can be used to understand $CO_2$ flux distributions and its fluctuations.

Traditional methods of univariate analysis cannot fully describe important physical states represented by data (Hoffman et al. 2011) because these methods neglect interactions between state variables. Regime identification has previously focused on



zonal geographic variations, related to the spatial distribution of light, temperature, mixing, and sea ice. Even the most sophisticated approaches (Fay and McKinley, 2014, Trochta et al., 2015) ignore the non-zonal, regional character of the ocean biogeochemistry and its interaction with physical circulation like in the western boundary current regions, in the upwelling zones on the eastern boundaries, and in the eddying field.

Cluster analysis is a highly effective multivariate analysis method for large, high frequency data sets because it can interpret the Earth System as a complex, geophysical network of nodes and edges, called a climate network (Peron et al., 2014). The nodes are spatial grid points and the edges represent statistical connectivity of a time series. Clustering seeks to identify the critical nodes and natural patterns of a climate network without any training or predetermined spatial-temporal guidelines,

therefore it is an unsupervised graph theory method (Jain, 2009; Phillips, 2015). The type of clustering applied here is k-means cluster analysis.

Cluster analysis has been successfully applied to various dynamical systems in order to extract the organized states of climate networks. For example, this technique has been used to define atmospheric weather states by identifying cloud regimes (Jakob

and Tselioudis, 2003; Rossow et al., 2005; Williams and Webb, 2009; Bodas-Salcedo et al., 2014; Oreopoulos et al., 2016). Bankert and Solbrig (2015) were able to extract a three-dimensional cloud representation from cluster analysis. This technique has also been used to characterize water types in lakes (Trochta et al., 2015), hydraulic habitat composition in rivers (Hugue et al., 2015), phenology patterns in forests (Mills et al., 2011), solar variability (Zagouras et al., 2013), ENSO phenomena (Radebach et al., 2013), and regions with characteristic hydrological response (Halverson and Fleming, 2014), among many

other applications. Beyond identifying regimes, cluster analysis can be useful in model assessment applications, like that of Wood et al. 2015, which used weather states derived from cluster analysis for process studies, satellite calibration, and model evaluation. To our knowledge, the ocean carbon cycle has not yet been evaluated using this technique.

The structure of the paper is as follows. Section 2 describes the data sets used in this study, both the observational sources as

well as the model experiments. These data are available on data.giss.nasa.gov/oceans/carbonstates/. Section 3 presents the k-means cluster analysis in observations and model output. Extended description of the methodology is provided in section 3.1 as we apply it to the North Atlantic basin for the derivation of the observations and model clusters (regimes/carbon states), the sensitivity analysis, and the attribution temporally and regionally of each regime. Section 3.2 repeats the analysis presented in Section 3.1, but now applied to the Southern Ocean. All the scripts used here are also available on

data/giss.nasa.gov/oceans/carbonstates/. Finally, general discussion and conclusions are provided in Section 4.



## 2 Data

### 2.1 Observations

**Air-Sea flux of $CO_2$ and $pCO_2$, surface wind speed, sea surface temperature and salinity**

The 12-month climatology of the air-sea flux is obtained from the Carbon Dioxide Information Analysis Center (LDEO

database (NDP-088); Takahashi et al., 2009). It is derived from the difference between surface water $pCO_2$ ($pCO_{2sw}$), air $pCO_2$, and the air-sea gas transfer rate. Surface water $pCO_2$ climatological mean distribution was obtained from 3 million measurements from 1970 to 2007, and normalized to a reference year 2000. The $pCO_2$ of the air is computed from the GlobalView $CO_2$ concentration zonal mean, NCAR monthly mean barometric pressure, SST, and salinity. Other variables in the data set that are pertinent to this analysis are wind speed (derived from the 1979-2005 climatological mean NCEP-DOE

AMIP-II Reanalysis wind speed field), climatological sea surface temperature (from NOAA Climate Diagnostic Center Objective Interpolation), and salinity (from the NODC World Ocean Database 1998). All variables are available as a 12-month climatology at a 4º x 5º resolution.

### Nitrate

The nitrate monthly climatology at 1 degree horizontal resolution is obtained from the World Ocean Atlas 2013 version 2

(Boyer et al., 2013). It is collected from in situ measurements at standard depth levels and is available as annual, seasonal, and monthly climatologies. Nitrate is an essential nutrient that limits the growth of phytoplankton, which are responsible for fixating carbon dioxide from the atmosphere. Therefore, $pCO_2$ levels in the surface ocean depend partially on the abundance of nitrate.

### 2.2 Numerical Simulations

The NASA-GISS modelE2.0 output used for this analysis comes from 5 ensemble model simulations of the 20[th] Century with realistic forcing as used in CMIP5 experiments. The model physics is somewhat different than the modelE2 used in the CMIP5 experiments mostly due to improved representation of the ocean mesoscale mixing. The physical ocean and the biogeochemistry modules are described in detail in Romanou et al (2013; 2017). Briefly here we note that the ocean model is a non-Boussinesq mass-conserving ocean model with 32 vertical levels and 1x1.25° horizontal resolution. The vertical

coordinate is a stretched z-level coordinate and has a free surface and natural surface boundary fluxes of freshwater and heat that are obtained by the atmospheric model. In addition to advection and turbulent mixing, it also includes a scheme for isopycnal eddy fluxes and isopycnal thickness diffusion. The interactive ocean carbon cycle model consists of a biogeochemical model (NASA Ocean Biogeochemistry Model (NOBM) Gregg and Casey, 2007) and a gas exchange parameterization for the computation of the $CO_2$ flux between the ocean and the atmosphere (Romanou et al., 2013). NOBM

utilizes ocean temperature and salinity, mixed layer depth and the ocean circulation fields, and the horizontal advection and vertical mixing schemes obtained from the host ocean model as well as shortwave radiation (direct and diffuse) and surface



wind speed obtained from the atmospheric model to produce horizontal and vertical distributions of several biogeochemical constituents. The carbon submodel parameterizes the cycling of carbon through the phytoplankton, herbivore and detrital components, affecting the dissolved inorganic and organic carbon in the ocean and interacting with the atmosphere. The air–sea gas transfer of $CO_2$ is parameterized following Eq. (1). Atmospheric $pCO_2$ ($pCO_{2atm}$) is the saturation concentration of $CO_2$

in equilibrium with a water–vapor-saturated atmosphere at a total atmospheric pressure P and a given atmospheric $pCO_2$ level:

$$pCO2_{atm} = \frac{P}{P^0} \, CO2^0 \tag{3}$$

where $P_0 = 1$ atm and $[CO_2]^0$ is the saturation concentration at 1 atm total pressure.

The gas transfer velocity is given by

$$kw = c(\frac{Sc}{660})^{-1/2} wspd^2 \tag{4}$$

where wspd is the surface wind speed and c is the piston velocity coefficient taken here equal to $0.337/(3.6 \times 10^5)$. The value of c has been agreed upon by the Ocean Carbon Model Intercomparison Project, phase II (OCMIP-II) so that the global, annual mean gas transfer coefficient for carbon dioxide (kw, $K_0$) is equal to 0.061 mol/m²/yr/l atm for preindustrial times. Sc, the Schmidt number, is computed using the temperature of the host ocean model following Wanninkhof (1992). The gas transfer velocity kw is computed only over open water. The solubility of $CO_2$ in the water $K_0$ is also parameterized based on OCMIP

using prognostic temperature, salinity and sea level pressure. In these model runs, the global average of the atmospheric concentration of $CO_2$ follows the Mauna Loa measurements (Le Quéré et al., 2015), although regionally atmospheric $CO_2$ is allowed to vary due to the distributions of the ocean sources and sinks.

The five ensemble member runs were averaged into one ensemble mean to account for the intrinsic climate variability that is

not adequately resolved in climate models of low spatial resolution. The model output for the years 1995 – 2005 was then averaged again to produce a 12-month climatology for the purpose of direct comparison with the observations.

The model output and the observational data were interpolated onto the same grid, which is the Takahashi ocean grid, and the ocean mask was conformed across all observational and model data sets. Because the Takahashi atlas contains mostly missing

values at high latitudes, we excluded these across all observational and model datasets. All data products are available in the Ocean Carbon States Database (data.giss.nasa.gov/oceans/carbonstates).

## 3. Methodology and Results

We will test the clustering method in the two main carbon source/sink basins of the global ocean (Takahashi et al., 2009), namely the North Atlantic (defined as 80°W to 45°E, 0° to 90°N) and the Southern Ocean (defined as 180°W to 180°E, 90°S

to 40°S). We apply k-means clustering analysis to SST and $pCO_{2sw}$ observational datasets (see Section 2.1) and to the NASA-



GISS climate model results in order to describe the ocean carbon regimes in the North Atlantic and the Southern Ocean, and use them to evaluate model biases in $pCO_{2sw}$.

## 3.1 North Atlantic

### 3.1.1 North Atlantic carbon states from observations

Probability density distributions show that both SST and $pCO_{2sw}$ exhibit a broad range of values in the basin of North Atlantic (see Fig. S1 in the Supplemental Material): $pCO_{2sw}$ values span 50 - 450 uatm and temperatures range between -2 and 30ºC. Based on these value ranges, we display the 12 monthly 2D histograms of $pCO_{2sw}$ and SST on Fig. 1. These show highest frequency of occurrence for pCO2sw values in the range of 300 to 400 uatm and temperatures in the range of 10 to 30ºC. Certain months (December, January, February and March) show a higher frequency of occurrence of cold temperatures (-2 to

$2^0$C) and low pCO2sw (50 to 300 uatm) than others. Fig. 1 also reveals certain patterns, for example, January – April share a similar S-shaped curve and no tilt and June – September exhibit a diagonal tilt that reflects a tendency for higher temperatures to collocate with higher $pCO_{2sw}$ values. We aim to mechanistically identify a small number of random (in space and time) situations of the pairs of $pCO_{2sw}$ and SST that may be able to describe all the 2D histograms in Figure 1. We will call those organized situations "ocean carbon states", or "regimes".

*(Figure 1)*

Since this is a few member dataset (we only have 12 months in the Takahashi climatology), a visual inspection is possible; however, histograms of larger datasets would not be as easily grouped together visually. A more objective method would

therefore be necessary and in the next few paragraphs we will test whether the k-means clustering analysis can play this role.

### k-Means clustering

The statistical method used here to determine the $pCO_2$-SST regimes in the North Atlantic is the k-means clustering method (Anderberg, 1973; Jacob and Tselioudis, 2003), which partitions and allocates the spatially and temporally defined 2D

histograms of $pCO_2$-SST (Fig. 1) into groups called clusters. This algorithm iteratively searches for a predefined number of clusters (k), stopping when the squared error between the mean of each cluster and the 2D histograms assigned per cluster is minimized (Jain, 2009). In the first phase of the analysis, centroid clusters (seeds) are randomly initialized for a certain number of data samples. Each histogram is then assigned to its nearest (in a Euclidean distance sense) cluster centroid for N number of iterations and then the centroid of the gaining cluster is recalculated, if doing so reduces the sum of distances from each

histogram to the centroid.



More than one iteration (N) is necessary to have convergent clustering results because each analysis initializes at a random cluster centroid. Multiple iterations allow for the analysis to find the minimum of the sum of squared Euclidean distances (between each histogram to cluster centroid). In order to ensure the reproducibility and consistency of results, we used an algorithm that plotted any change in the minimum as the number of iterations was increased (see Fig. S2). For our analysis,

the maximum number of iterations required to reach convergence was 10.

**Sensitivity to predefined number of clusters**

In applications of k-means clustering, the number of clusters k is predetermined. To ensure that the number of clusters chosen is representative of the system, typically, one needs to repeat the technique for various values of k, using visual analysis to

select the optimal value when there is no additional information contained in a new cluster. Objective methods have been tested (e.g. Bankert and Solbrig, 2015), where the average radius of each cluster (the distance from the centroid to the most distant sample within a cluster) is computed for decreasing k. They found that when the number of clusters falls below the optimal k, the average radius grows rapidly. We employ a similar methodology here where the optimal number of k is determined using a sensitivity test that compares the average distance of all monthly 2D histograms to the centroid of their

assigned cluster versus that of other clusters, using an assigned score which quantifies that distance. Negative or low values indicate poorly matched histograms that have the largest distances from the centroid in the cluster. The maximum score is 1 which indicates a perfect match. Since our data sets are 12-month climatologies, the maximum number of possible predetermined clusters (k) is 12. Then we run the scoring algorithm for k = 2 through k = 12, each time yielding 12 scores per k that represent how well each 2D histogram is matched with its representative cluster. As an example, Fig. 2a shows the score

plots for k = 2, k = 3, and k = 4. The 12 scores are then averaged for each test and are normalized by the number k (Fig. 2b). The optimal number of clusters is determined by identifying the k with the highest score and no significant change of the score thereafter. As shown in Fig. 2b, the average score tends to plateau as the number of clusters is increased. For the pair of variables $pCO_2$-SST in the North Atlantic, the optimal number of clusters k is 3.

*(Figure 2)*

Figure 3 depicts the regimes for k = 2, k = 3, and k = 4. When only two clusters are predefined, i.e. for k = 2, the first cluster (Regime 2A, Fig 3a) is dominated (30% of the time) by $pCO_2$-SST pairs in the ranges of 350 – 400 uatm and 25-30ºC. The second cluster (Regime 2B) is dominated (20%) by $pCO_{2sw}$ values within 300 – 350 uatm and 350 – 400 uatm and SST values

in the ranges -2-20ºC and 25-30ºC, respectively. When we choose more clusters initially, i.e. as k is increased to 3, Regime 3B is very similar to Regime 2A and Regime 3C is analogous to Regime 2B. Regime 3A is a new state that was unresolved in k = 2. For k = 4, Regime 4B and 4C appear to be almost equivalent, and both derived from Regime 3B, which probably indicates that there is no new information gained by requiring 4 clusters. Similar visual inspection of the results for k > 4





confirms our objective analysis result that k = 3 is the optimal number of clusters in the pCO₂-SST space in the North Atlantic basin.

*(Figure 3)*

**Temporal attribution for each cluster**

In order to characterize the states of the ocean carbon system yielded by cluster analysis in the previous section, we perform a temporal attribution analysis by determining when each cluster occurs.

The k-means analysis provides the distance of each member 2D histogram to the centroid of the cluster it belongs, and by

identifying which 2D monthly histogram is closest to the centroid we are able to associate its cluster with certain months of the climatology. Figure 4 shows that in the North Atlantic basin, regime 1 is represented by months January, February, March, and April and we call this the "winter regime"; regime 2 occurs during June, July, August, September and October and we will thus call it the "summer regime"; regime 3 occurs in May, November, and December and we will call this the "transition regime" because it reflects a mixed season in between the winter and summer regimes. Because our domain is the entire North

Atlantic, from the equator to the poles, the seasons do not necessarily correspond to boreal seasons commonly known as winter/summer/fall/spring but at least the winter and summer regimes encompass the corresponding boreal seasons, hence are named after them.

*(Figure 4)*

**3.1.2 North Atlantic carbon states in models**

Next we perform clustering analysis on the model data. As described in Section 2.2, we construct the ensemble mean climatology for the period 1995-2005 for all the relevant fields (pCO₂SW, SST, SSS, surface wind speed, and nitrate) from 5 simulations of Earth's historical climate of the 20th Century performed with the NASA-GISS climate model.

Using the same sensitivity test as in the observations, the optimal number of clusters for the model is also found to be 3 (Fig. S3). The clusters themselves are similar to the observations (Fig. 5) in that the same bins of most likely values are identified but with somewhat different frequencies of occurrence. Direct comparison between the clusters is possible because temporal attribution on the model regimes leads to similar results as in the observations (Table 1; Fig. S4): the model winter regime corresponds to months December, January, February, March, and April; the summer regime corresponds to months July,

August, September, October, and November; the transition regime corresponds only to June. Table 1 shows that the model winter regime is broader than the observed by two months (December and May) while the model summer regime is lagging by one month (starts in July, while in the observations starts in June).

*(Table 1)*

As an example of comparison between the temporal regimes, for the winter regime both the model and the observations show that the dominant pairs are in the range of 300 – 350 uatm and 20 – 25C, at 30% and 25% relative frequency. However, other weaker pairs are not well represented in the model, e.g. the range 50 – 200 uatm and -2 – 10ºC.

In order to better understand the model-observation cluster discrepancies shown in Fig. 5 we seek to identify where in the North Atlantic basin these regimes develop. To do so, the frequencies of occurrence associated with each $pCO_2$-SST bin of the regimes in Fig. 5 are averaged over the months in each regime and mapped on the North Atlantic basin. The regional attribution for the observations (Fig. 6a) is compared with the model regional attribution (Fig. 6b).

Using Fig. 5, we know that the dominant range of values for the observations and the model winter regime occurs for values 300 – 350 uatm and 10 – 20ºC. These ranges of the variable pair are found in the subtropical North Atlantic (see Fig. 6 and Fig. S5), but with higher frequency in the model. The observations also show greater frequency of occurrence of the value pairs in the subpolar region (values 50 – 350 uatm and -2 – 10ºC, identified in Fig. 6 and Fig. S5). In contrast, the dominant range of $pCO_2$-SST pairs in the summer regime occurs in the tropics (values 350 – 400 uatm and 25 – 30ºC; Fig. 6 and Fig. S5) and is of higher frequency in the observations than the model. The transition regime shows a mix of the winter and summer regimes for both observations and model. The model results in this regime indicate higher frequency in the subpolar region than in the observations.

Since the model regimes are different enough from the observations, particularly in spatial distribution, all subsequent analysis will be performed with both observations and model output composited on the observational regimes. In other words, the model output will be averaged over those months that define each observational cluster in Fig. 4.

*(Figure 5)*
*(Figure 6)*

### 3.1.3 Model air-sea flux of $CO_2$ error analysis and bias attribution

The ocean carbon states identified in the previous section can provide a framework for model assessment against the observations. In this section we seek to identify biases in the simulated flux of $CO_2$ and attribute them to leading biases in physical and biogeochemical processes.

Figure 7 depicts the air-sea flux of $CO_2$ composited over the observed regimes in both observations (Fig. 7a) and model output (Fig. 7b). In the winter regime, model outgassing (in shades of blue) is confined only to the tropics, whereas in the observations



there is also a tongue of outgassing at about 60ºN. Similarly, in the transition regime, the model has more extended uptake region in the subpolar North Atlantic than the observations. While the summer regime is better represented in the model than the other two regimes, all three regimes show that model uptake is stronger than in observations at mid and high latitudes and outgassing is also stronger in the model in mid to low latitudes.

*(Figure 7)*

To trace the source of model biases implied in Figure 7 we need a better understanding of the physical or biogeochemical processes that control the air-sea flux of $CO_2$ in the model. Using the clustering analysis results from the previous section, we
can investigate the underlying processes that might be responsible for the bias.

The process attribution is performed using a Taylor expansion of the model bias as shown in Eq. (5). The model flux bias, $\Delta F$, depends on the biases of $pCO_{2SW}$, SST, SSS and wind speed (wspd) such that:

$$\Delta F \sim \frac{\partial F}{\partial pCO2_{SW}} \Delta pCO2_{SW} + \frac{\partial F}{\partial SST} \Delta SST + \frac{\partial F}{\partial SSS} \Delta SSS + \frac{\partial F}{\partial wspd} \Delta wspd \tag{5}$$

where $\Delta q$ is the bias of the variable q, defined as the root mean squared error (RMSE) between the observations and the model. $\frac{\partial F}{\partial q}$ is a weight term that represents dependence of the flux on that variable and q={ $pCO_{2SW}$, SST, SSS, wspd} and is determined by the slope of a linear fit in the scatter plot of the flux F with each variable q for each carbon state. Since the North Atlantic basin is both zonally and meridionally broadly extended and because the carbon states' regional distribution
(Fig. 6) is quite complex, we identify certain regions where the linear fits will be more appropriate approximations of the {F,q} relationships. As shown in Fig. S5, we identify a subpolar region (-2 to 10ºC, 50 to 350 uatm), a subtropical region (10 to 20ºC, 300 to 350 uatm), and a tropical region (20 to 30ºC, 300 to 400 uatm). Results of the regional scatter plots and the linear fit for each regime are shown in Fig. S6 and are synthesized in Figure 8. Each contribution term (summation term in Eq. 5) is calculated from the multiplication of the weights and the RMSEs. Figure 8 then shows that over most of the North Atlantic,
the flux biases are attributed mainly to errors in the $pCO_{2SW,}$ although in subpolar regions other terms such as salinity biases and wind speed biases become important. It therefore makes sense to further investigate biases in $pCO_{2SW}$ and the processes these are attributed to, as presented in Eq. (2).

*(Figure 8)*

Similarly to Eq. (5):



$$\Delta pCO2_{SW} \sim \frac{\partial pCO2_{SW}}{\partial SST}\Delta SST + \frac{\partial pCO2_{SW}}{\partial SSS}\Delta SSS + \frac{\partial pCO2_{SW}}{\partial WSPD}\Delta WSPD + \frac{\partial pCO2_{SW}}{\partial NITRATE}\Delta NIT \qquad (6)$$

We perform the Taylor expansion of the bias for each of the regimes that we computed, calculating the weights and RMSEs in the same way as described for $CO_2$ flux biases. The estimates of the linear fit slopes of the scatter plots are shown in Fig. S7 and the composites of the contributions in Fig. 9.

Overall Fig. 9 shows that the biases in the subpolar region are larger than anywhere else in the North Atlantic basin, as the contribution to the bias in pCO2sw are an order of magnitude larger there. Specifically, in the subpolar region, wind speed biases emerge as responsible for the winter and transition regime biases in pCO$_2$sw while salinity biases dominate the summer bias in pCO$_2$sw. In the winter and transition regimes, the quasi-cyclonic subpolar gyre, driven by energetic winds and wind-outbreaks, leads to Ekman divergence in the surface layer that controls the pCO2sw biases near the coast. At the same time, winter-time convective mixing is responsible for biases in the strength of the Meridional Overturning Circulation that are known to influence open ocean pCO2sw (Romanou et al, 2017). In the summer regime, GISS model sea-ice concentration is higher than observed hence melting will lead to significant surface salinity biases. Inaccurate model representation of the magnitude and fluctuations of the cyclonic wind stress curl as well as the sea-ice retreat and associated salinity changes are probably responsible for deficient physical characterization of the model ocean circulation, which would result in misrepresentations of the pCO$_2$sw and thus the $CO_2$ flux in the model.

In the subtropics, nitrate is found to be the largest contributor for the winter regime biases, wind speed is the main contributor in the summer, and salinity is the main contributor for the transition regime. The subtropics are characterized at the surface by anticyclonic circulation and a strong western boundary current, the Gulf Stream. Gyre subduction supports downwelling which brings nutrients and pCO$_2$ to depth. Nitrate utilization by ocean biology during the winter regime is probably inaccurate in the model while wind speed biases are known to be larger in the summer than the winter regime in the model.

In the tropics, biases in wind speed, nitrate and salinity are again found to be important. Here, nitrate biases are probably due to misrepresentation of nitrogen fixation in the GISS climate model, whereas wind speed and salinity biases are associated with well-known biases in the intensity and position of the Inter Tropical Convergence Zone (ITCZ) that controls cloudiness, temperature gradient, and rainfall. The ITCZ moves north in the summer and south in the winter, therefore a wind speed bias in the transition regime in the model could be explained by an inaccurate model reproduction of how the ITCZ affects the wind during its transitional movement. The ITCZ increases precipitation, thus decreasing salinity, therefore how salinity changes by season as a result of the shifting ITCZ could explain the winter regime bias.

*(Figure 9)*



### 3.2 Southern Ocean

### 3.2.1 Southern Ocean carbon states based on observations and model simulations

For the Southern Ocean, probability density distributions show that both SST and $pCO_{2SW}$ exhibit a broad range of values (see Fig. S8). Temperatures range between -3 and 20°C whereas $pCO_{2SW}$ values span 20 - 400 uatm. The 12-monthly 2D histograms

of $pCO_{2SW}$ and SST in the Takahashi dataset are shown in Fig. S9.

Fig. 10a shows the scores of each regime for k=2, 3, and 4 and Fig. 10b provides the results of the sensitivity test for the Southern Ocean. These demonstrate that k = 3 is the optimal choice for the predetermined number of clusters k, despite having a few poor scoring points as seen in the middle panel of Fig. 10a. The histogram that has yielded a negative score corresponds

to the November 2D histogram. An increase to k = 4 improves the match of the November histogram, but then May has a very low score. Both November and May correspond to mixed-season phases and the inability of the cluster analysis to match these months well at k = 3 or k = 4 may suggest that the domain chosen, the entire Southern Ocean, is too large in spatial coverage and too diverse in conditions to be well represented at once. Nevertheless, an optimal value of k = 3 is chosen based on the sensitivity test and a subjective, visual inspection of apparent patterns in the 12-monthly 2D histograms. A value of k = 3 is

also chosen for the model analysis based on Fig. S10 which shows that this is also the optimal number of clusters.

*(Figure 10)*

Temporal attribution, which is estimated using the method described in section 3.1.1, is shown in Fig. S11 and the results for

both the model and the observations are summarized in Table 2. Note that all subsequent analysis considers the austral seasons when referring to "winter" and "summer". The temporal attribution shows that the observations and model data are clustered in regimes that correspond to almost the same months. The only difference is that November is accounted for in the transition regime for the observations as opposed to the winter regime for the model. It is noted, however, that November is technically a "poorly matched" 2D histogram in the observations cluster routine (Fig 10a).

*(Table 2)*

In order to explain the model and observed regimes (Fig. 11), the frequencies of occurrence of each bin in the cluster are mapped onto the Southern Ocean (Fig. 12). Regional analysis within the basin is defined using Fig. S12, which leads to the

following zonal definitions based on ranges of $pCO_{2SW}$ and SST: coastal Antarctic divergence zone (SST -3 – 3°C and $pCO_{2SW}$ 20 – 250 uatm), Antarctic convergence zone (SST 3 – 10°C and $pCO_{2SW}$ 250 – 400 uatm), and subtropical convergence zone (SST 10 – 20°C and $pCO_{2SW}$ 250 – 400 uatm). Despite the strong temporal attribution agreement between the model and the observations, the regional attributions show much less correspondence. For example, in the summer regime, the observations





show the highest frequency of occurrence between 250 – 350 uatm and 10 – 20ºC along the subtropical convergence zone (roughly along 40ºS) while the highest frequency of occurrence for the model is for the pair 250 – 350 uatm and 0 – 5ºC, occurring in the coastal region (poleward of the divergence zone along 60ºS). The winter regime corresponds only within 20 – 150 uatm and -3 – 10ºC along the coast. Higher pCO2sw and SST values vary in frequency of occurrence and spatial distribution. Within 250 – 350 uatm, the model transitional regime has higher frequency for SST between 0 – 10ºC whereas the observations show higher frequency for 10 – 20ºC. Also, the model transitional regime spatially delineates its highest frequencies of occurrence strongly within the Antarctic convergence zone whereas the observations demonstrate a more heterogeneous distribution of pCO2sw-SST frequencies.

*(Figure 11)*
*(Figure 12)*

Significant discrepancies between observations (Fig. 13a) and model (Fig. 13b) CO2 flux composites on the regimes obtained from observations motivate our subsequent error bias analysis. For example, we see that in the summer regime, outgassing in the model is restricted to the subtropical convergence zone whereas outgassing in the observations occurs in small patches at different latitudes and longitudes. Also, in the model summer regime, uptake is much stronger, confined to the coast and occurring in a much wider area than in the observations. In the winter regime, the entire model basin is a sink for CO2 whereas in the observations there is a zonally confined outgassing belt south of 50ºS and an uptake belt north of it. The transition regime shares a mix of the same discrepancies as in the summer and winter regimes.

*(Figure 13)*

Bias attribution is computed for the three zonally-defined regions indicated in Fig. S12. Based on the bias computations in Eq. (5) for pCO2sw, SST, salinity, and wind speed with respect to CO2 flux, pCO2sw is again shown to be the driving variable in most of the flux biases in the Southern Ocean (Fig. S13; Fig. S14 for scatter plots). We therefore seek to understand the processes that control the pCO2 biases in the model, using the Taylor expansion in Eq. (6) (Fig.14; Fig. S15 for scatter plots).

For almost all regimes and regions, biases in nitrate are large partly because of lack of a closed, state-of-the art nitrogen cycle representation in the climate model. However, observations are too scarce in the region, due to inclement weather and biases to specific seasons, so the model skill would be more adequately assessed as more in situ measurements are made (e.g. from the SOCCOM experiment; Johnson et al., 2017). Nevertheless, the model underestimates surface nitrates in the Southern Ocean in particular because of a large nitrate deficit in the subsurface ocean which upwells in the subantarctic zone and flows into the Antarctic Circumpolar Current region. This is related to processes such as denitrification and accurate remineralization in the deep ocean. SST is the second-most dominating variable for biases in the coastal Antarctic. Inspection of the model



biases shows that south of 70ºS the model water column is colder than in observations, hence upwelling there will bring colder waters near the surface. Interestingly, surface salinity biases are relatively very important in the region south of the subtropical convergence zone which suggests that a study of water mass formation in that region needs to be used to explain the biases.

*(Figure 14)*

**Data availability**

Data and analysis scripts can be accessed at www.data.giss.nasa.gov/oceans/carbonstates.

**4 Conclusions**

This proof-of-concept study establishes the ability of k-means cluster analysis to find realistic regimes in observational data
of the ocean carbon cycle. A method is provided for an objective way to accurately determine the optimal number of clusters for the cluster analysis. The study also explores how to attribute the cluster outputs temporally and spatially in order to explain the regimes in a spatial-temporal framework. Composites of the $CO_2$ flux and a quantitative exploration of the effect of each field on $pCO_{2SW}$ bias is also demonstrated.

Direct comparison between model and observations serves to validate the model and the error analysis can explain any model biases that need to be corrected. Applying k-means clustering analysis in two main regions of the world that are known to be critical for the global ocean carbon cycle, namely the North Atlantic region and the Southern Ocean, defines the priorities for model improvement: in the North Atlantic biases in surface salinity, wind speed and surface temperature whereas in the Southern Ocean priorities are nitrate and surface salinity. Clearly the GISS climate model would benefit from more realistic
representation of the nitrogen cycle in the ocean as a whole.

The goal of this study is to enable us to apply this methodology to big data, in order to find the interannual and regional patterns in larger, higher frequency climate networks. Cluster analysis will be tested on smaller regions as well. Other variable pairs besides $pCO_2$ and SST will also be explored, such as $CO_2$ flux and chlorophyll, in order to assess other drivers in Eq. (1).
Finally, higher order clustering and classification techniques will be analyzed in order to determine the most efficient and successful method.

All routines and datasets used in this study are freely available in the Ocean Carbon States page of the NASA-Goddard Institute for Space Studies web portal (data.giss.nasa.gov/oceans/carbonstates).



**Competing interests**

The authors declare that they have no conflict of interest.

**Acknowledgements**

Resources supporting this work were provided by the NASA High-End Computing (HEC) Program through the NASA
Center for Climate Simulation (NCCS) at Goddard Space Flight Center.
Funding was provided by NASA-ROSES Modeling, Analysis and Prediction 2013 NNX14AB99A-MAP for GISS Model-E
development and NNX15AJ05A NASA Cooperative Agreement 2015-2018.
Data used to generate figures, graphs, plots, as well as analysis were archived at NCCS dirac repository, numerical codes are
maintained and archived at GISS and all data and codes are available upon request from A. Romanou. Clustering analysis was
performed using the MATLAB ver 2015 computing environment. The authors wish to thank Robert Schmunk for his help in
setting up the Zenodo page and the GISS portal.

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

**Tables:**

|  | Winter Regime | Summer Regime | Transition Regime |
|---|---|---|---|
| Observations | 1 – 4 | 6 – 10 | 5, 11, 12 |
| Model | 12 – 5 | 7 – 11 | 6 |

30 **Table 1: Comparison of the monthly attributions (months numbered 1 – 12) for model and observations in the North Atlantic.**



|  | Winter Regime | Summer Regime | Transition Regime |
|---|---|---|---|
| **Observations** | 6 – 10 | 1 – 3 | 4, 5, 11, 12 |
| **Model** | 6 – 11 | 1 – 3 | 4, 5, 12 |

Table 2: Comparison of the monthly attributions (months numbered 1 - 12) for model and observations in the Southern Ocean. The regimes are named following the austral seasons.





**Figures:**

**Figure 1: Monthly 2D histograms of partial pressure of CO₂ in the surface water (pCO₂sw) and sea surface temperature (SST) in the North Atlantic (defined as 80°W to 45°E, 0° to 90°N) from the Takahashi observational dataset. The horizontal axis is pCO₂sw (uatm) and the vertical axis is SST (°C). The bin interval is 15 uatm and 1.6°C. The colorbar describes the actual frequency of occurrence of each bin.**



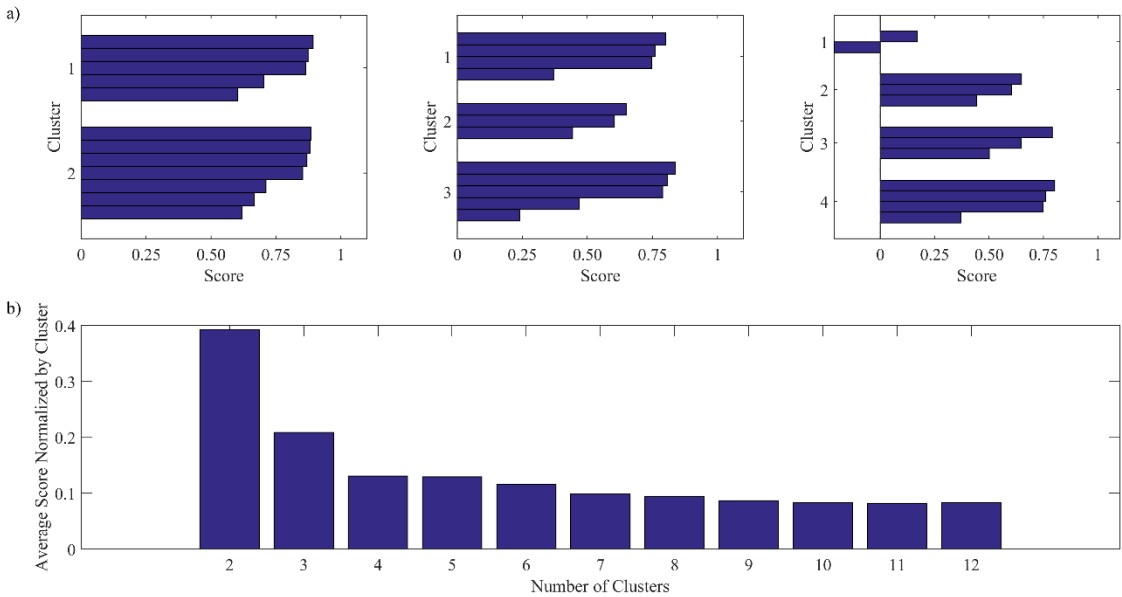

**Figure 2: a) Scores for each cluster analysis of observational data in the North Atlantic for k = 2, k = 3, k = 4, where k is the predetermined number of clusters. b) Average scores for each cluster analysis with increasing k, normalized by the number of clusters k.**

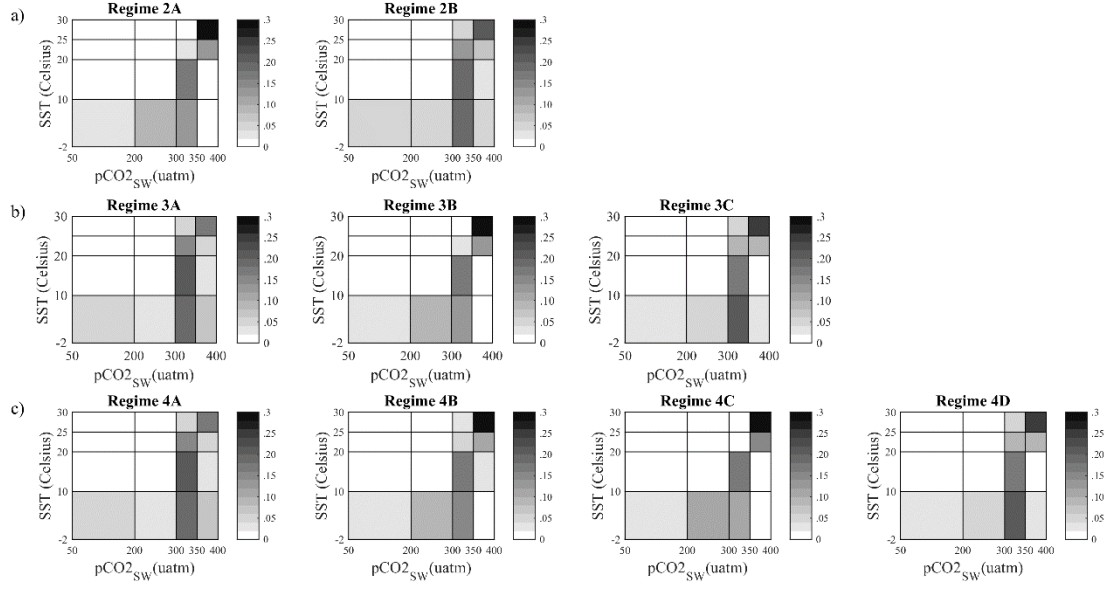

**Figure 3: Cluster analysis output (regimes) for a) k = 2, b) k =3, and c) k = 4 for the North Atlantic, from the Takahashi observational dataset. Frequencies of occurrence are divided by total number of frequencies per regime in order to show fraction of total.**





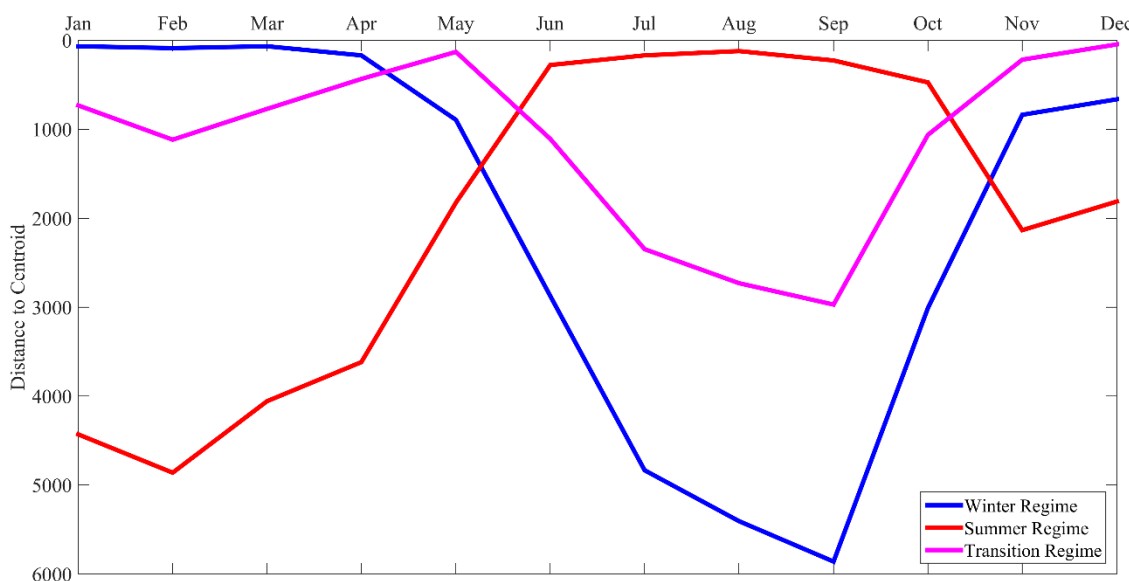

**Figure 4: Monthly attribution of each ocean carbon regime in the Takahashi observational dataset. Temporal attribution is based on the distance of each monthly 2D histogram to the centroid of each cluster.**

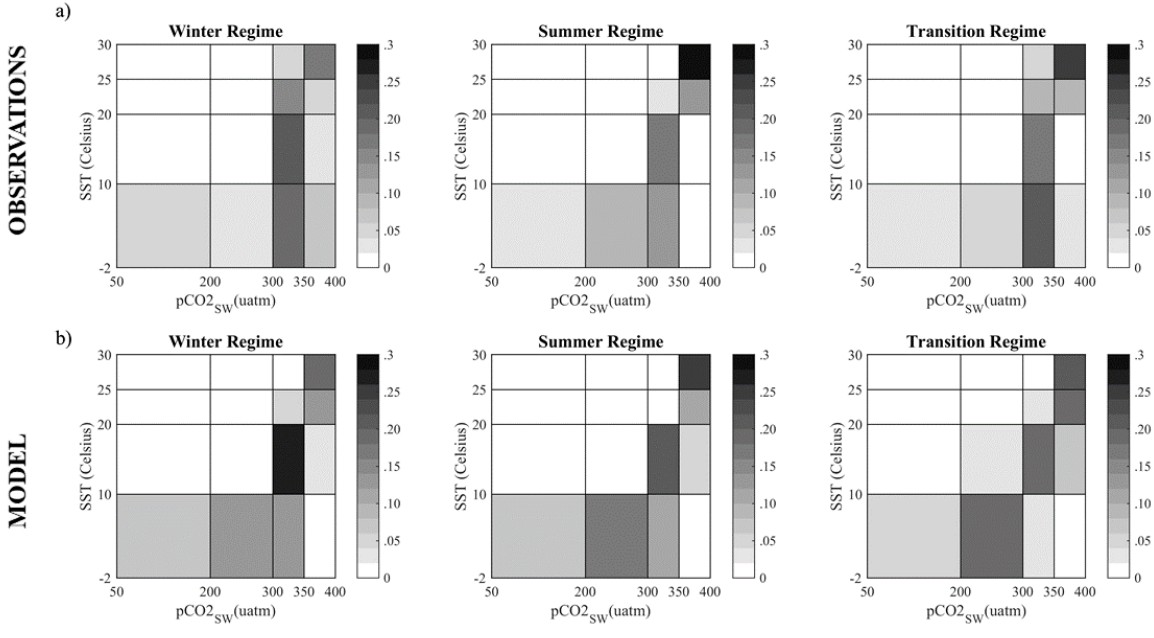

5    **Figure 5: Ocean carbon states (regimes) in (a) the observations and (b) the model output.**





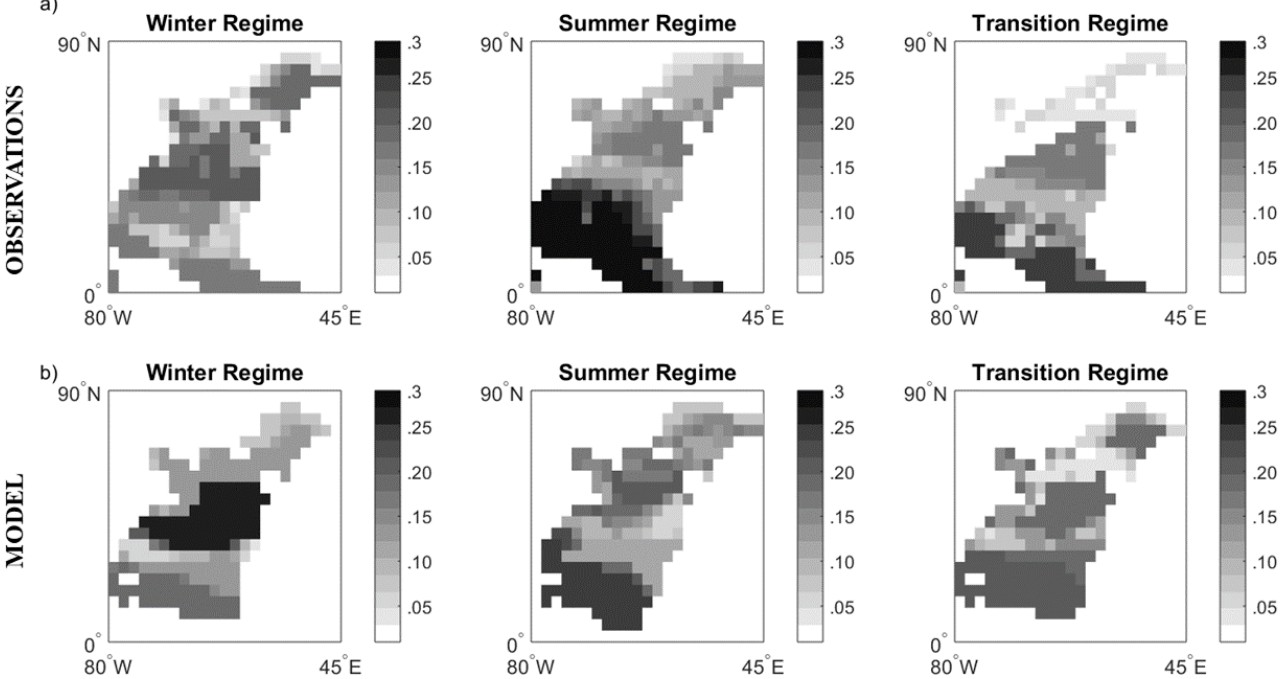

**Figure 6: Regional attribution of each regime depicted in Fig. 5 in (a) the observations and (b) the model simulations. The frequencies of occurrence of each bin (value pair) in the clusters of Fig. 5 is mapped onto the North Atlantic grid.**





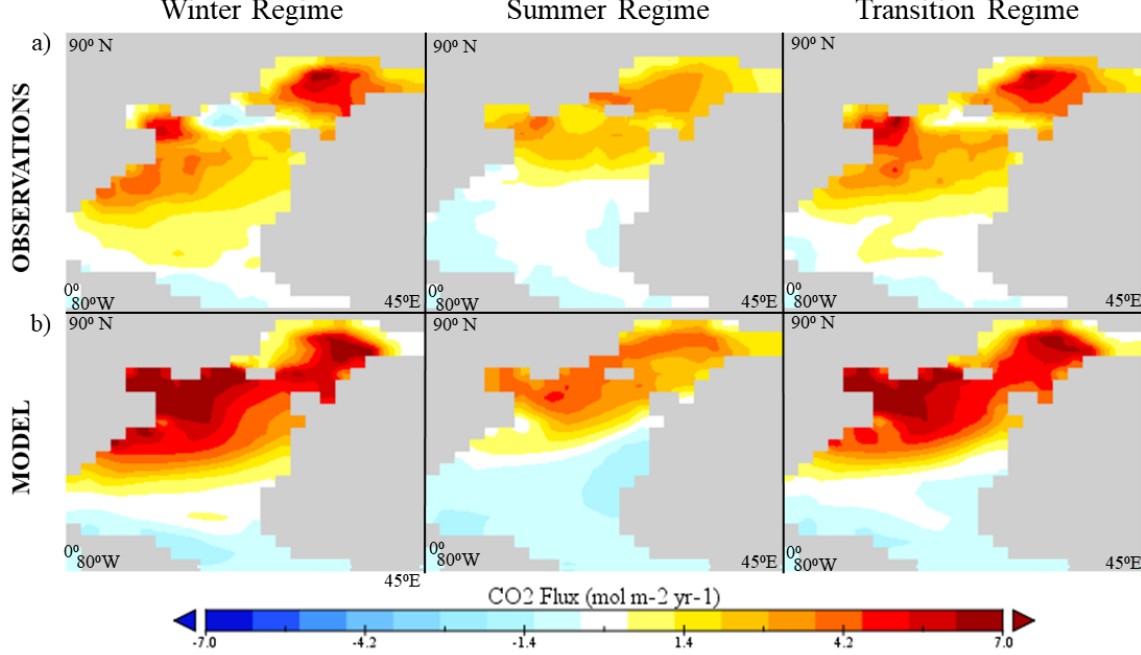

**Figure 7: Composites of the CO₂ Flux field over the observed regimes for (a) the observations and (b) the model. Both the observations and the model data are composited over the same months as determined by the temporal attribution of the observation data set, shown in Fig. 4. Blue shades indicate outgassing, red shades indicate uptake.**




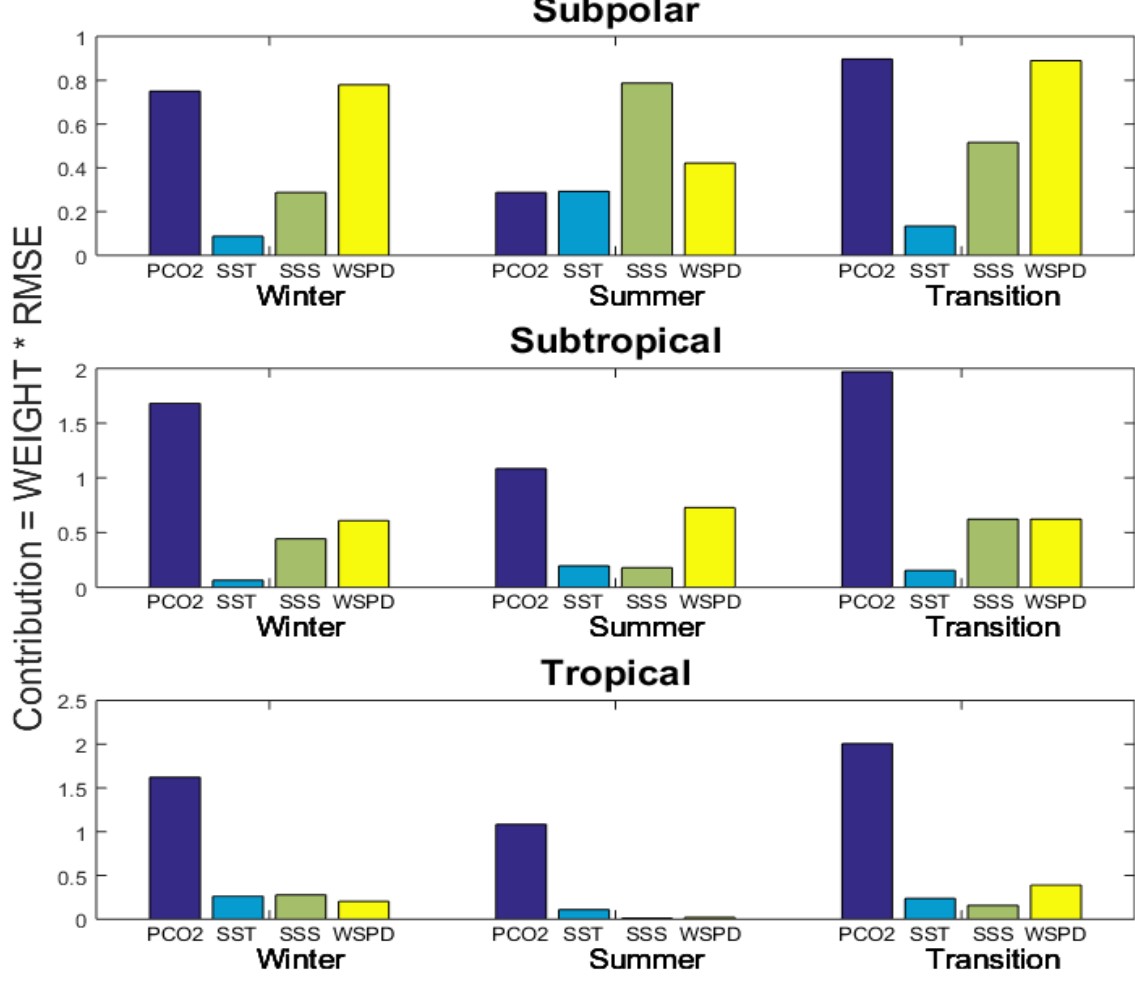

**Figure 8: Contributions of each of the variables pCO₂, SST, SSS, wspd to the overall air-sea flux $\Delta F$ bias. The contributions are computed as the products of the weights and the RMSEs of each variable q as described in Eq. (5). See text for detailed explanation.**







**Figure 9: Contributions to the pCO₂sw bias in the model from SST, SSS, wind speed and nitrate in the winter, summer, and transitional regimes. Contributions are computed as in Fig. 8 (see details in the text). The entire North Atlantic is differentiated into subpolar, subtropical, and tropical regions to better account for regional differences in the model biases and obtain better linear fit for the computation of the weights in Eq. 6.**





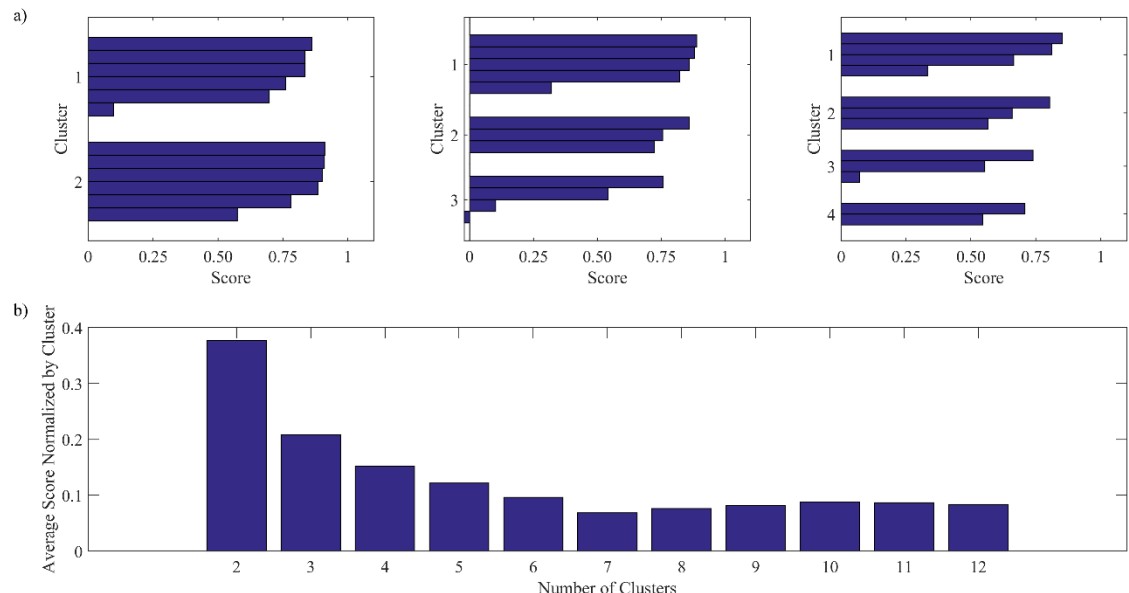

**Figure 10: a) Scores for each cluster analysis for k = 2, k = 3, k = 4 for the observational data in the Southern Ocean. b) Average scores of each clustering analysis for increasing k.**

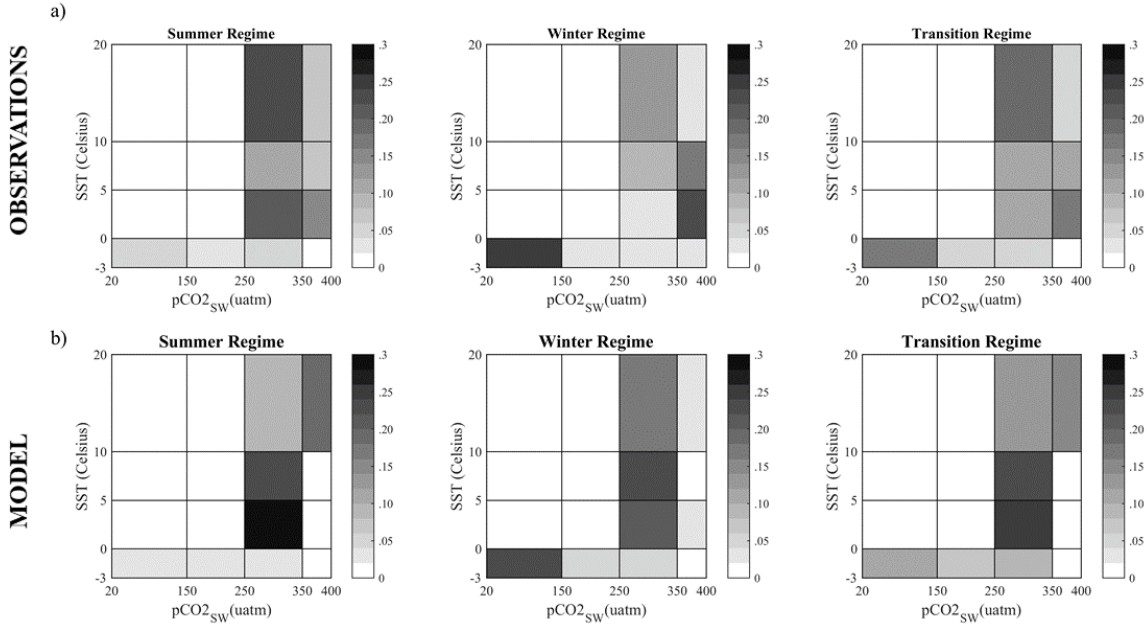

5    **Figure 11: Comparison between the regimes in (a) the observations and (b) the model output.**



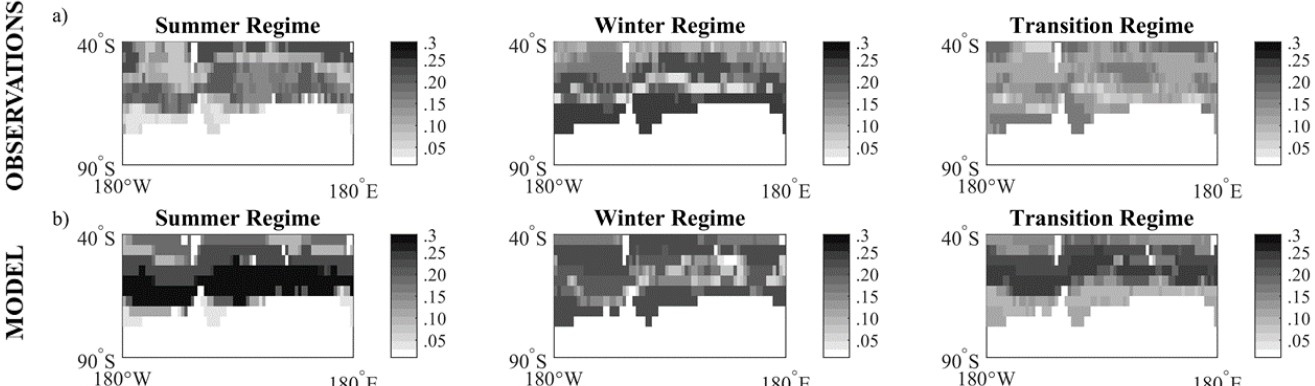

**Figure 12: Regional attribution of each regime in (a) the observations and (b) the model simulations. Each spatial grid point for every month is associated with its relative frequency of occurrence in the cluster output, and then the months are averaged per regime to output the average frequency of occurrence in each regime. Model regimes are calculated using the monthly groups identified by the observation temporal attribution.**

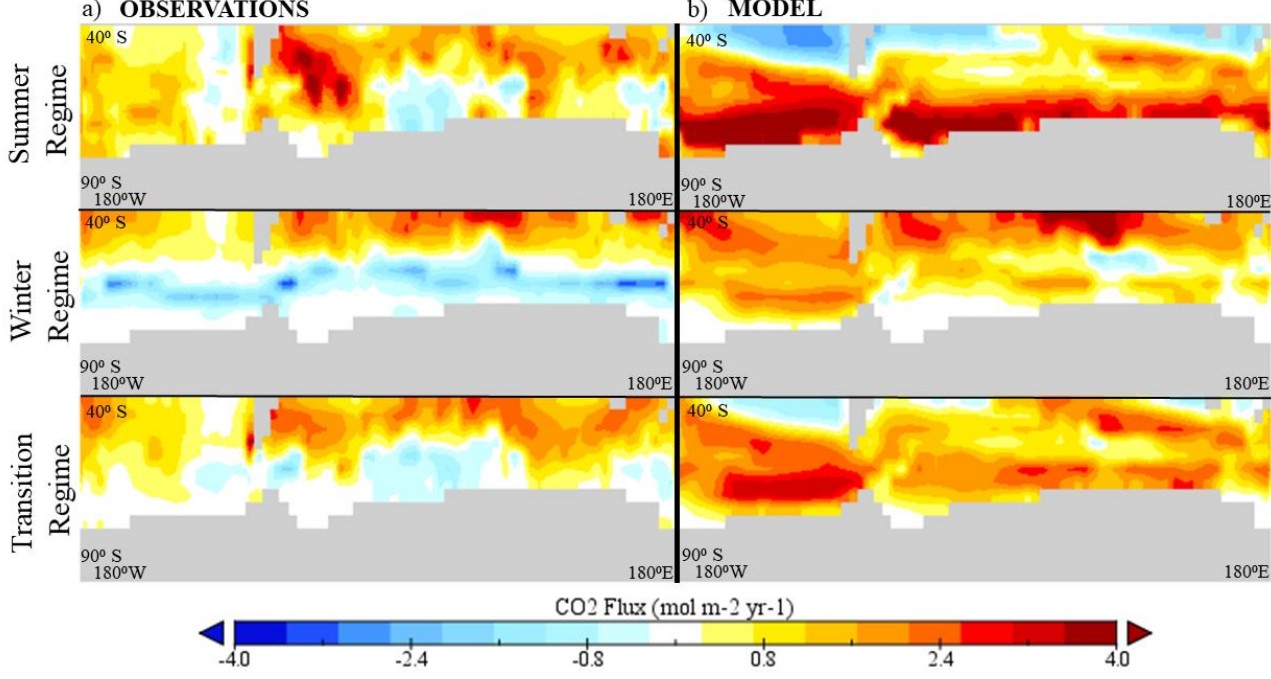

**Figure 13: Composites of the $CO_2$ Flux field over the observed regimes for (a) the observations and (b) the model. Both the observations and the model data are composited over the same months as determined by the temporal attribution of the observation data set. Blue shades indicate outgassing, red shades indicate uptake.**



**Figure 14: Contributions to the pCO₂sw bias in the model from SST, SSS, wind speed and nitrate in the winter, summer, and transitional regimes. The Southern Ocean is differentiated into the coastal Antarctic divergence zone (roughly polewards of 60ºS), Antarctic convergence zone (roughly 60ºS-50ºS), and subtropical convergence zone (roughly 50ºS-40ºS) to better account for regional differences in the model biases and obtain better linear fit for the computation of the weights in Eq. 6 (see Fig. S12).**