# Peer review of "The "Ocean Carbon States" Database: a proof-of-concept application of cluster analysis in the ocean carbon cycle"

_Earth System Science Data, 2017_

## Referee Comment (RC1) · Anonymous Referee #1 · 28 Nov 2017

In this study, the authors use a technique called k-means clustering to delineate oceanic regions based on SST and pCO2 data (and model). They then use these domains to look for the likely sources of model biases on pCO2 fluxes in these clusters. They conclude that wind speed is a major culprit in the North Atlantic, while a faulty biology (as diagnosed by nitrate) is likely the main source of error in the Southern Ocean.

As currently written, it is unclear what advantages k-means clustering provide over other methods. It seems the main accomplishment from using this method was to split the North Atlantic into 3 zones, which the authors refer to as the tropical, subtropical

and subpolar regions, and to identify winter from summer, with transition periods in-between. This is hardly news.

The arguments presented for choosing the number of clusters, i.e k=3 in this case, are also very arm-wavy. The number of clusters seems like it could vary as a function of the bin size used in the histograms, a binning which is very coarse, and is a function of subjective decisions by the authors.

The paper contains also too many figures, 14 in the main text, and 15 in the supplement. Why some figures are put in the main text while others in the supplement is not totally clear from the presentation, though. It is also not clear at all what the main result is.

Overall, I would not recommend publication of this manuscript in its current state. The paper is difficult to understand, the methodology not clear and I am left confused as to what the main point was and the merit of the method is. A comprehensive rethink of the manuscript is, in my view, necessary.

Specific comments

P1, L9: pattern

P2, L22: column.

P2, L32: "Traditional methods of univariate analysis...". What are these methods exactly? Not sure what you are referring to here. K-means clustering can also be applied to a single variable.

P3, L6: I think kmeans clustering can also be applied to a single variable.

P3, L30: the actual address was https://data.giss.nasa.gov/oceans/carbonstates/

P6, L5: "in the North Atlantic basin"

P7, L6-p8-2: I find this explanation confusing and the conclusions of that section rather

unconvincing and not objective at all. Based on what I understood from this section and fig 2b, I would select k=6. Also, the argument for picking k=3 because "Regime 4B and 4C appear to be almost equivalent" (L32) does not seem to hold as to me, regime 2A and 2B are also "almost equivalent". It is a mystery to me what sets of rules the authors us to select their k. Also, the bin sizes of the histograms of Fig 2 are very large, spanning 5 degrees in temperature and 50uatm for the narrowest bins. What warrants this broad binning of the results in Fig 2? Presumably, the number of clusters k depends on the resolution of the data histogram (bin size). Finer bin sizes are able to pick out more patterns and so a higher k would be warranted. It is also not clear on what data the clustering algorithm is applied. Is it on each month independently or on all the data, or on some annual average?

P8, L15-17: "the seasons do not correspond to boreal seasons". This is a strange concept. How useful is it to pull together an entire region, define clusters, but then ignore well-known events such as the Spring bloom, or spring restratification and sea ice melt. This also raises the question as to which region in the North Atlantic is the most variable and which region dominates the pCO2 and SST variability. What is gained in this analysis by removing the geographical aspect? What is the main conclusion of that section? Could it be that clusters identified in July have nothing to do with the clusters identified in February? Given that some features may dominate different regions at different times, does it even make time to link clusters in time? The only role of clusters is to minimized a statistical measure of misfit, how do you guarantee a logical link between clusters in time?

P8, L26: Contrary to what is stated here, it doesn't look like the clusters in the observations and in the model look very much alike. The magnitudes of the color bar is quite different between the two.

P8, L27-27: "the same bins of the most likely values are identified...". This seems to be the case in all the analyses so far. Not clear how this statement can be used to justify the previous statement that obs and model agree in that context.

P8, L28: Comparison between Fig 4 and Fig S4 shows the cluster variability is totally different between the two cases. Is the main conclusion here simply that there are seasons in the data and there are seasons in the model?

P9, L21-23: not clear what you mean by "composited". Also the rationale for doing this totally eludes me. What is gained/lost from doing this?

P9, whole section 3.1.3: What is gained from this analysis that is not achievable simply by looking at the observed and simulated $CO_2$ fluxes? What information does k-means clustering contribute here? It would seem that a similar analysis done on each grid box would result in a much better and detailed analysis than if the domain is first split into various domains.

P12: same issues here in the Southern ocean as above for the North Atlantic about the arguments for choosing k=3.

Figures: Fig 1: poor choice of color scheme. Hard to distinguish between dark blue/light blue. Are all dark blue squares zero or non-zero?

Fig 8-9: Since $pCO_2$ is calculated from other variables, including SST, SSS, WSPD and NIT, what is the point of performing the analysis in two steps (i.e. fig 8)? It seems that Fig 9 simply shows that biases in SST, SSS and WSPD dominate the full bias, since $pCO_2$ is just a function of these variables

---

## Referee Comment (RC2) · Anonymous Referee #2 · 12 Dec 2017

**Review of Latto et al: The "Ocean Carbon States" Database: a proof-of-concept application of cluster analysis in the ocean carbon cycle. Submitted to ESSD**

**Summary:**

Latto and Romanou present a cluster analysis in the North Atlantic and Southern Ocean using the K-means clustering approach. Using the analysis, the authors identify a subset of biogeochemical regimes, or as they call it "ocean carbon states". The authors combine the Takahashi et al sea surface pCO2 with NOAA OI SST to identify these carbon states in the North Atlantic and the Southern Ocean to then compare the clustering results with an ocean model. Using this comparison, the authors asses the model performance and identify model bias.

**Strengths:**

The authors combine both observation-based estimates with models. Using a statistical method, the authors provide a rather novel way to identify model biases.

**Weaknesses:**

Unfortunately, there are many things in the current manuscript that are misleading, need better discussion and would benefit from substantial revision. I will list them here from the most to the least concerning:

- K-means Method: The authors use k-means without data normalization. As the authors state, e.g. in the North Atlantic, the range of pCO2 is 50-450µatm whereas SST ranges from 2-30°C, i.e. the pCO2 range is an order of magnitude larger then SST, therefore, when distances are computed in "the Euclidian distance sense" the results will be biased towards the pCO2. The authors will need to provide some evidence that this is not problematic, discuss why it is favorable to bias towards pCO2 or to normalize the data first in order to give SST and pCO2 equal weight.

- Terminology: The authors ignore that the observation-based pCO2 and SST products are based on statistical interpolation methods as well. E.G the Takahashi climatology is created by interpolating observations using an advection-based interpolation algorithm, whereas the SST is interpolated using an optimal interpolation method. Therefore, the products are (a) NOT OBSERVATIONS as claimed in the text but OBSERVATION-BASED products and (b) they come with their own uncertainty. It is therefore questionable, given the data sparsity in the Southern Ocean e.g. to use the Takahashi preduct as "ground truth" (also given that in the Takahashi et al 2009 paper the authors themselves calculate a global flux uncertainty of 50%). This needs to be discussed instead of wrongly assuming observation-based product=observations.

- Discussion of method/parameter choice: Nowhere in the text it is properly discussed why pCO2 and SST are chosen, and why the reader should accept these proxies as representatives for processes in both ocean basins. Despite this, on page 2 line 29 the authors claim that pCO2 and SST are independent variables, which is just wrong. pCO2 is certainly not independent from SST. As Takahashi et

al 1993 and 2002 show, a change of 1°C in temperature results in a 4% pCO2 change due to the solubility effect.

- The authors present many figures, but provide too little explanation about their meaning. E.g. what is largely missing is a discussion on potential fields these clusters can be applied to.

-  Introduction: The introduction is confusing rather than helping the reader build up the topic. The authors jump from paragraph to paragraph which to me seem to be very disconnected at times (e.g. paragraph 1 broadly discusses global warming, paragraph 2 jumps to gas exchange and paragraph 3 jumps to numerical simulations)

- Literature: I was disappointed that the authors missed to mention the already existing effort in using clustering techniques regarding the sea surface pCO2. Many studies (e.g. Lefevre et al 2005, Telszewski et al 2009, Sasse et al 2013, Landschützer et al 2013, 2014, Nakaoka et al 2013) use a self-organizing map (SOM) technique to build clusters in the surface ocean. Certainly, the aims of these other studies diverge from this one and certainly there are differences between AI methods (such as SOM) and K-means (despite the mathematical differences being actually very small), but nevertheless, the authors claim on page 3 that "To our knowledge, the ocean carbon cycle has not yet been evaluated using this technique" which might certainly hold true, but it behooves the authors well to at least discuss similar approaches to connect to the wider literature out there that indeed has applies similar methods for a similar purpose.

**Recommendation:**

While I value the effort and I certainly see the advantage of the technique and the resulting analysis, I believe the authors need to address the issues raised above before the manuscript can be considered for publication. I therefor **recommend at least major revisions of the manuscript**.

**Specific and minor comments to the text:**

Page 1 lines 21: "realistic, dynamical regimes" – I don't think the authors have shown anywhere that the regimes are "realistic"

Page 2 lines 24-31: More discussion is needed here.  Furthermore, there is no citation backing the text.

Page 3 lines 1-4: I am confused here. I am familiar with the Fay and McKinley 2014 identification (not the Trochta et al 2015), and to the extent of my knowledge they do not "ignore the non-zonal, regional character of ocean biogeochemistry". Please explain. As it currently reads the statement is wrong.

Page 4 and following: Observation-based products, as the climatologies presented use observations and usually a statistical interpolation algorithm to fill data gaps in space

and time. Therefore, the final climatology cannot be called observation anymore, but rather observation-based!!!

Page 5 line 10: I suppose the authors mean wind speed at 10 meter height rather than surface wind speed. Most gas transfer estimates are based on the 10-meter wind speed (such as the used Wanninkhof 1992 formulation)

Page 5 line 13: The Wanninkhof 1992 formulation is outdated as also highlighted by the author in several following, more recent publications.

Page 5 line 16: The reference to Le Quéré et al 2015 should be replace with the original data reference (Dlugokencky and Tans 2014). The global carbon budget combines all measurements/estimates for the budget, but individual contributions, such as the atmospheric CO2 should be acknowledged when used (this is also noted on page 1 of the excel sheet provided by the Global Carbon Project).

Page 5 line 23: Please mention that the Takahashi grid is a simply 4x5 degree regular grid

Page 5 line 25: The Takahashi estimate excludes the arctic ocean north of 80N

Page 6: k-means clustering: Firstly, I think this would fit better in section 2. Secondly, the authors do not provide sufficient explanation: E.G. it is not clear to everyone how euklidian distances are calculated. Therefore, it is easy to miss that the authors actually bias towards pCO2 (see major comment). Other terms not explained include "centroid clusters", "gaining cluster" and "seeds". These are abstract terms that need to be understood by the readers. Understanding a method means trusting a method!

Conclusions: line 10-11: "accurately determine the optimal number of clusters for the cluster analysis" - I disagree given the methodological caveats raised above.

Conclusions lines 15-20: I cannot follow why the authors conclude that biases in salinity temperature and wind are responsible for the mismatch in the NA and nutrients as well as salinity is responsible for the SO mismatch. Firstly, this result is for this model only. E.G. Lenton et al. 2013 have shown that there is large disagreements in models even with regards to the seasonality in CO2 and the drivers of all sorts of variability. Secondly, given the uncertainty from the observation-based estimate I am not convinced this conclusion is solid.

**References used in this review:**

Lefèvre, N., Watson, A. J., and Watson, A. R.: A comparison of multiple regression and neural network techniques for mapping in situ pCO2 data, Tellus, 57B, 375–384, 2005

Telszewski, M., Chazottes, A., Schuster, U., Watson, A. J., Moulin, C., Bakker, D. C. E., González-Dávila, M., Johannessen, T., Körtzinger, A., Lüger, H., Olsen, A., Omar, A., Padin, X. A., Ríos, A. F., Steinhoff, T., Santana-Casiano, M., Wallace, D. W. R., and Wanninkhof, R.: Estimating the monthly pCO2 distribution in the North Atlantic using a self-organizing neural network, Biogeosciences, 6, 1405–1421, doi:10.5194/bg-6-1405- 2009, 2009

Sasse, T. P., McNeil, B. I., and Abramowitz, G.: A new constraint on global air-sea CO2 fluxes using bottle carbon data, Geophys. Res. Lett., 40, 1594–1599, doi:10.1002/grl.50342, 2013.

Landschützer, P., N. Gruber, D. C. E. Bakker, U. Schuster, S. Nakaoka, M. R. Payne, T. Sasse, and J. Zeng: A neural network-based estimate of the seasonal to inter-annual variability of the Atlantic Ocean carbon sink, Biogeosciences, 10, 7793–7815, doi:10.5194/bg-10-7793-2013, 2014

Landschützer, P., N. Gruber, D. C. E. Bakker, and U. Schuster: Recent variability of the global ocean carbon sink, Global Biogeochem. Cycles, 28, 927–949, doi:10.1002/2014GB004853, 2014

Nakaoka, S., Telszewski, M., Nojiri, Y., Yasunaka, S., Miyazaki, C., Mukai, H., and Usui, N.: Estimating temporal and spatial variation of ocean surface pCO2 in the North Pacific using a selforganizing map neural network technique, Biogeosciences, 10, 6093–6106, doi:10.5194/bg-10-6093-2013, 2013.

Takahashi, T., J. Olafson, J. Goddard, D. Chipman, and S. Sutherland:, Seasonal variations of CO2 and nutrients in the high-latitude surface oceans: A comparative study, Global Biogeochem. Cycles, 7(4), 843–878, 1993

Takahashi, T., et al.: Global sea-air CO2 flux based on climatological surface ocean pCO2, and seasonal biological and temperature effects, Deep-Sea Res. II, 49, 1601–1622, 2002

---

## Referee Comment (RC3) · Anonymous Referee #3 · 13 Dec 2017

**Review of Latto and Romanou: The "Ocean Carbon States" Database: a proof-of-concept application of cluster analysis in the ocean carbon cycle:**

**Summary:** This study presents a methodology for the analysis of model output against observations of $CO_2$ fluxes. The methodology is based on a "data-mining" technique, where both observations and model output are subjected to a cluster analysis, which identifies the main seasonal regimes of variability of sea surface temperature (SST) and surface water $pCO_2$ in the North Atlantic, and Southern Ocean. The study is focused on the evaluation of uncertainties related to the ocean component driving the $CO_2$ flux in the model.

Specifically, the analysis is initially carried out on a monthly climatology of sea surface temperature (SST) and surface water $pCO_2$ observations, in order to identify the number of clusters (K) that best represent variations in the data. This analysis is repeated with model output. Subsequently, model and observations maps of $CO_2$ flux for each regimen/cluster (K) are compared in order to discern regions of disagreement. Finally, the authors identify sources of bias in the model by evaluating the physicochemical factors that contribute the most to the disagreement between model and observations.

**Major scientific comments:** This study test an interesting way of assessing uncertainties in climate models. A key step in the analyses is the determination of the number of clusters K, which defines the main regimes within observations and model output. My main concern here is the apparent subjectivity is this determination, described in "**Sensitivity to predefined number of clusters**". The optimal number of clusters is determined *"...by identifying the K with the highest score and no significant change of the score thereafter."* (page 7, line 21). This is based on a visual inspection of Fig.2b, 10b, S3b, and S10b. Regarding the case for the North Atlantic, the authors mention that the optimal K number is 3 for both observations and model output. Based alone on the visual assessment of Fig. S3b, I am inclined to say that 5 is a more appropriate number of K for model outputs. How would this discrepancy affect your results? Furthermore, it seems convenient that in both cases, North Atlantic and Southern Ocean, the optimal number of K in model and observations is the same, how would you compare model and observations if they had a different number of K?

Overall, these data-mining technique should provide a more objective method for model assessment. However, the subjectivity involved in the determination of the number of K, seems to defeat this purpose.

**General comments:**

-   Abstract: This is a problematic section. There is a lot of emphasis on the utility of data-mining techniques and little or none mention of the biogeochemical findings regarding model biases in the North Atlantic or the Southern Ocean. It is not clear what are the main findings/conclusions of the study.

- Methods, Section 3.1.1: It took me several reading attempts to understand the way in which the K-means clustering works. It may be useful to include a diagram or include equations describing the iterative clustering process. As mentioned above, the method used to determine the final number of K clusters needs to be improved, or at least, better justified.

- Conclusions: *"A method is provided for an objective way to accurately determine the optimal number of clusters for the cluster analysis."* I highly disagree with this statement based on my **Major scientific comments**, above.

- The discussion of the attribution of model errors is well written. However, the conclusions are compromised by the subjective determination of the number of K.

- Figures. The distribution of figures between the main manuscript and the supplementary material is confusing. I recommend merging the frequency histograms in figures S1 and S8 in one figure, and including it in the main manuscript. In general figures are duplicated for the North Atlantic and Southern Ocean. With the exception of the 2D histograms, I recommend merging some of the similar figures for different regions (e.g., Fig 2 and Fig 10.), and also include in the main manuscript the analysis of the Southern Ocean (Fig S9, S10, S11, and S13). Fig S2 is not clear. It seems that some of the lines in the label are not included in the figure (e.g., K=2 and K=9 are both dark blue, but there is only one black blue line in the figure, and they would be impossible to distinguish). Fig 2a, 10a, 3a, and 10a: For more clarity, it would help to label the month corresponding to each bar.

**Specific comments:**

Page 10, Line 21: *"As shown in Fig. S5, we identify a subpolar region…"* What is the basis for the determination of these sub-regions in the North Atlantic and Southern Ocean? If they are arbitrarily chosen, "*we define*", would be a more appropriate wording. The criteria used to define these regions is important in the final attribution of processes driving the errors between models and observations.

Page 11, Line 25: "*Here, nitrate biases are probably due to misrepresentation of nitrogen fixation in the GISS climate model…*" In oligotrophic regions, relative errors in nitrate may be higher as the absolute concentration is overall lower, with much less seasonal amplitude when compared with higher latitudes. I agree that nitrogen fixation is likely a problem, but it might be useful to consider this aspect as well (see Arteaga et al, 2015, GRL, doi:10.1002/2014GL062937).

Page 12, Line 29: "*For almost all regimes and regions, biases in nitrate are large partly because of lack of a closed, state-of-the art nitrogen cycle representation in the climate model. However, observations are too scarce in the region, due to inclement weather and biases to specific seasons, so the model skill would be more adequately assessed as more in situ measurements are made (e.g. from the SOCCOM experiment; Johnson et al., 2017).*" This is also problematic regarding observations of $CO_2$ flux, particularly in winter. How do you account for a possible bias towards summer fluxes due to a lower amount of observations during winter months?

**Final remarks:** Overall, I find this study interesting and important for the community. However, further clarity is needed in the explanation and justification of the methods involved. My impression is that this study could be considered for publication after major revisions.

---

## Author Comment (AC1) · 2 Feb 2018

We wish to thank the reviewer for their thoughtful comments and helpful suggestions. Please find the following attached: 1) A point-by-point reply to each of the comments made by the referee (including any useful references and figures), 2) Revised manuscript and supplementary materials (clean versions), 3) Revised manuscript and supplementary materials (showing tracked changes from initial submitted manuscript to revised manuscript).

Please also note the supplement to this comment:

[Figure]

https://www.earth-syst-sci-data-discuss.net/essd-2017-113/essd-2017-113-AC1-supplement.zip